# Hierarchical communication of chirality for aromatic oligoamide sequences

Jiajia Zhang[1], Dan Luo[1], Chunmiao Ma[1], Lu Huang[1] & Quan Gan [1✉]

The communication of chirality at a molecular and supramolecular level is the fundamental feature capable of transmitting and amplifying chirality information. Yet, the limitation of one-step communication mode in many artificial systems has precluded the ability of further processing the chirality information. Here, we report the chirality communication of aromatic oligoamide sequences within the interpenetrated helicate architecture in a hierarchical manner, specifically, the communication is manipulated by three sequential steps: (i) coordination, (ii) concentration, and (iii) ion stimulus. Such approach enables the information to be implemented progressively and reversibly to different levels. Furthermore, the chiral information on the side chains can be accumulated and transferred to the helical backbones of the sequences, resulting in that one of ten possible diastereoisomers of the interpenetrated helicate is finally selected. The circular dichroism experiments with a mixture of chiral and achiral ligands demonstrate a cooperative behavior of these communications, leading to amplification of chiral information.

---

[1] Hubei Key Laboratory of Bioinorganic Chemistry & Materia Medica, Hubei Engineering Research Center for Biomaterials and Medical Protective Materials, School of Chemistry and Chemical Engineering, Huazhong University of Science and Technology, Wuhan, P. R. China. ✉email: ganquan@hust.edu.cn

The communication of chirality is one of the fundamental features of living systems, such as signal transduction processes and enzyme asymmetric reactions[1]. The fabrications of artificial entities with chiral communication property, which enables the individual chiral components to adopt coincident chiral conformations and leads to accumulation and amplification of overall chiral sense, brings about attractive prospects to the field of chemistry and materials[2–4]. Conventionally, the chiral communication can mainly occur when the individual chiral components are spatial approach[5,6]. Either covalent or noncovalent coupling strategies, such as metal coordination[7–10], π–π stacking[11–17], hydrogen bond[18], guest template[19–22], and making use of structural folding[23–26], have been explored to huddle the individual components together for communication. Despite this versatility, most artificial systems with chirality communication reported so far are of unadjustable structure, aiming to reach compact communication of chirality. This device, however, makes the communication immobile and thus loses the potential for processing and handling of chirality information. Therefore, to develop chirality communication systems with dynamic and reversible controllability is highly desired[27].

The hierarchical self-assembly is emerging as a powerful means to create large functional complexes. The programmed manner is sensitive to the changes in internal parameters or external stimuli, which may also allow the associated functions to be operated step by step along with the successive assembly[28,29]. In the present study, we show that the chirality of the aromatic oligoamide sequences can be progressively communicated and regulated in a hierarchical manner. The chirality of aromatic amide sequences stems from the twist of amide: amide segments can distort and deviate the neighboring aromatic rings from co-planarity giving rise to the $P$ or $M$ helical conformation of the molecular strands. And the anticipation of feasible control this helical handedness by use of coordination comes from our and others' recent findings[30,31] that these sequences adopted coincident helical conformations when assembled to the $M_2L_4$ metallasupramolecular complex. This observation implies the twisting of aromatic oligoamides can be coupled under the metal coordination

connection, i.e., a situation in which it is energetically favorable for sequences of one helical conformation to have neighbors of the same helicity. This phenomenon is also coincident with some coordination complexes of aliphatic peptide ligands, the helicity of which can be communicated and amplified[7–10]. These studies also hinted a fact that the control of the helical chirality is highly dependent on the restriction of the steric crowding of sequences. In view of this, we proposed that the helicity integration of the aromatic amide ligands might not only present in a single helicate but would also be more easier to meet within an interpenetrated helicate architecture[32–38], since the two monomeric helicates in this interlocked framework appears to be more sterically restricted from twisting. Furthermore, we envisaged that the chirality communication could be even strengthened when the interpenetrated helicate was contracted by trapping Cl⁻ anions in the cavities. Such hierarchical strategy therefore enables the communication to be broken up into several sequential levels, and programming the chirality information by the corresponding controls is feasible.

## Results

**Design and self-assembly of helicates**. Aromatic amide pentamer as a ligand **L** was synthesized by multistep condensation reactions (Supplementary Scheme 1). Two pyridine groups were introduced to the two extremities for coordination. The long structure and curvature of the skeleton make it possible to assemble into an interpenetrated helicate under coordination with $Pd^{2+}$ (Fig. 1), as it is comparable to the ligands known for the constructed interpenetrated cages that were developed by the Kuroda and Clever groups[32–38].

A mixture of monomeric helicate **1** and dimeric helicate **2** was formed by heating **L** with $[Pd(CH_3CN)_4](BF_4)_2$ (0.5 eq.) in $CD_3CN$ for 12 h at 56 °C, as supported by two sets of signals observed via $^1H$ NMR spectroscopy with full assignments (Fig. 2a–c) and high-resolution ESI mass spectrometry (Supplementary Figs. 2–5). The proportions of the monomeric and dimeric helicates varied with concentration as shown by $^1H$ NMR

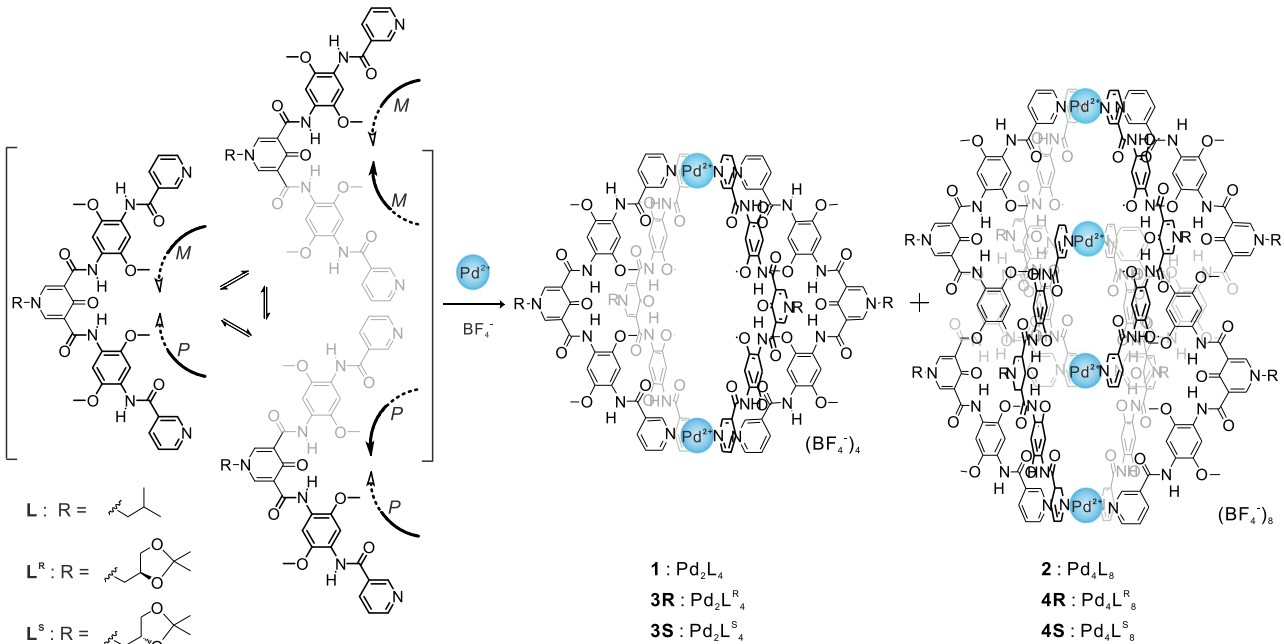

**Fig. 1 Synthesis of the monomeric and dimeric helicates.** The reaction of the corresponding ligand (2 equiv.) with $Pd(BF_4)_2$ (1 equiv.) led to the formation of the mixtures of the monomeric and dimeric helicates.

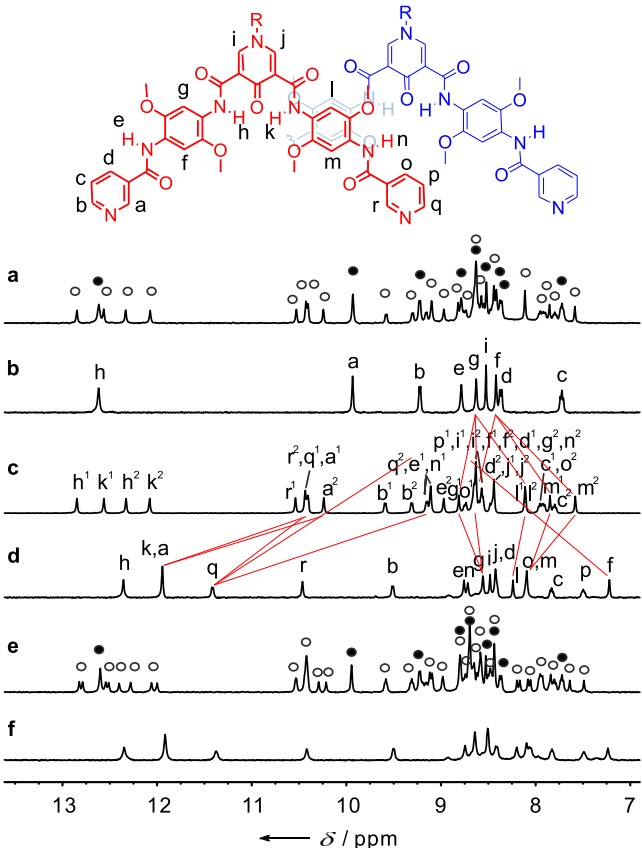

**Fig. 2 $^1$H NMR (600 MHz) spectra of the helicates and the corresponding Cl$^-$ complexes in CD$_3$CN (500 μL). a** Excerpts of $^1$H NMR spectra of mixture of **1** and **2** (ca. [**L**] = 1.8 mM) at equilibrium. **b 1** formed from coordination of **L** and Pd$^{2+}$ at low concentration (ca. [**L**] = 0.2 mM). **c 2** formed from coordination of **L** and Pd$^{2+}$ at high concentration (ca. [**L**] = 8.0 mM). **d** Excerpts of $^1$H NMR spectra of [2Cl⊂**2**]. **e** Excerpts of $^1$H NMR spectra of **3** S and **4** S (ca. [**L**] = 1.8 mM) at equilibrium. **f**, Excerpts of $^1$H NMR spectra of [2Cl⊂**4** S]. The Cl$^-$ complexes (**d**, **f**) were formed by addition of NBu$_4$Cl 0.5 mmol to the correspongding helicate solutions (**a**, **e**), respectively. Amide and aromatic signals of the monomeric helicates and of the dimeric helicates are marked with filled and empty circles, respectively.

spectra (Supplementary Fig. 7), suggesting that they are in equilibrium in the solution. The dimerization constant was found to be $1.65 \times 10^4$ M$^{-1}$ by the integration of the NMR signals. The diffusion-ordered $^1$H NMR spectroscopy (DOSY) measurement further supported that these two species have distinct diffusion coefficients, which were calculated to be $4.90 \times 10^{-10}$ and $4.15 \times 10^{-10}$ m$^2$ s$^{-1}$ (with hydrodynamic radii of 1.2 nm and 1.4 nm), respectively (Supplementary Fig. 8). The observation of the nuclear Overhauser effect (NOE) cross peaks between the signals of methoxyl (–OCH$_3$) and pyridine protons (r) shed light on its interlocked architecture of **2** (Supplementary Fig. 12): correlations between these protons are unlikely to be intramolecular and are not observed for the single ligand or monomeric helicate **1**, but may emerge from an intermolecular stacking of the ligands with a slippage motif, as supposed in the interlocked structure.

**Chirality selection from the monomeric to dimeric helicate.** The $^1$H NMR spectra show that the monomeric helicate **1** only possesses one signal for each of the ligand's proton environments (Fig. 2b). Whereas based on our prior experience[30], two structural isomers (a meso-structure with *PM* helicity and a race-complex

with *PP* and *MM* helicities, where helicity is defined by the orientation of each Pd$^{2+}$ cation with respect to the center of the ligand, see Fig. 1 and Supplementary Fig. 13) are expected for **1**, considering the twisting of the amide moieties and given the consistency of twisting in one helicate. Low temperature NMR studies show neither significant chemical shift nor signal splitting, inferring the isomers of **1** are in fast exchange on NMR timescale (Fig. 3b). The dimeric helicate **2**, by contrast, manifests four times of the NMR signals with equal intensities (Fig. 2c), suggesting that there are two thermodynamically stable diastereoisomers of **2** in slow exchange. We inferred that this isomerization might also be caused by the amide twisting. The *P/M* helical interconversion mechanism of the aromatic amide sequences is known to take place either via an untwisted or a partially unfolded conformation[39,40], both of which need the sequence to undergo a conformational stretching state. This stretching motion during the helicity interconversion would be more restricted in the confined interlocking helicate than in the monomeric helicate, thus resulting in the slower dynamics of twisting for **2**.

With regarding to the assembly of dimeric helicates from two monomeric ones, it requires dynamically disassociation and reassociation of ligand-metal linkage. This process may be inhibited by using more kinetically inert metal during the assembly. In fact, none of the dimeric helicate but the monomeric one **1**-Pt$^{2+}$ was formed when Pt$^{2+}$ was employed in a control experiment (Supplementary Fig. 16). Moreover, in contrast to the monomeric helicate **1**, the isomers of the Pt$^{2+}$ analogous complex display slower exchange, as two sets of signals were observed by NMR. The proportion of these isomers (*PM* vs *PP*/*MM*) could be biased by the encapsulation of anion guests, addition of either Cl$^-$ or SO$_4^{2-}$ anion would result in one of each isomers being preferred (Supplementary Fig. 17), the behavior of which has also been founded for the similar metal complex in our previous research[30]. The result also supports our hypothesis that the metal coordination can correlate and unify the helicity of the ligands within a helicate structure, because there would be more complicate NMR signals for the helicate if the twisting of each ligands was not consistent.

As mentioned above, we supposed that the amide twisting would be more restricted in the dimeric helicate with the approach of the ligands to each other, and that may lead to the selection of helicity. In principle, ten kinds of isomers can be considered as interpenetrated helicates derived from the amide twisting orientation and depicted as *PPPP*, *MMMM*, *PPPM*, *MMMP*, *PPMP*, *MMPM*, *PMMP*, *MPPM*, *PPMM*, and *PMPM*, respectively, according to the defined helicity to the single helicate. By reference to the crystal structures available for the hydrogen-bonded aromatic amide helicates[30,31], the energy-minimized calculations[41] (Gauss'09, DFT, B3LYP/LANL2DZ, see Supplementary molecular modelling section for details) of **2** revealed that dimerization could only occur to two monomeric helicates with the same helicity rather than with the heterogenous helicity probably due to the unfavorable steric hindrance caused by mismatching assembly. Two pairs of the interpenetrated helicate enantiomers (*PPPP*/*MMMM*, and *PMMP*/*MPPM*) were visualized as the most energetically minimized structures (Fig. 3c and Supplementary Table 1), which is in agreement with the NMR experiments that the number of signals of **2** is twice as large as those of other reported interpenetrated coordination cages where no twisting occurs[32–38,42–46]. We posited that the aromatic π–π stacking also plays a role in this chirality selection. To gain the maximum of π–π interaction it requires the ligands with the same twisting at the intertwined section, thus only the dimeric helicates with *PPPP*/*MMMM* and *PMMP*/*MPPM* helicities were favored. In the modelling structure of **2**, a large overlapping area of ligands was observed between the phenyl rings at the intertwined section (Supplementary Fig. 18), and it is supported

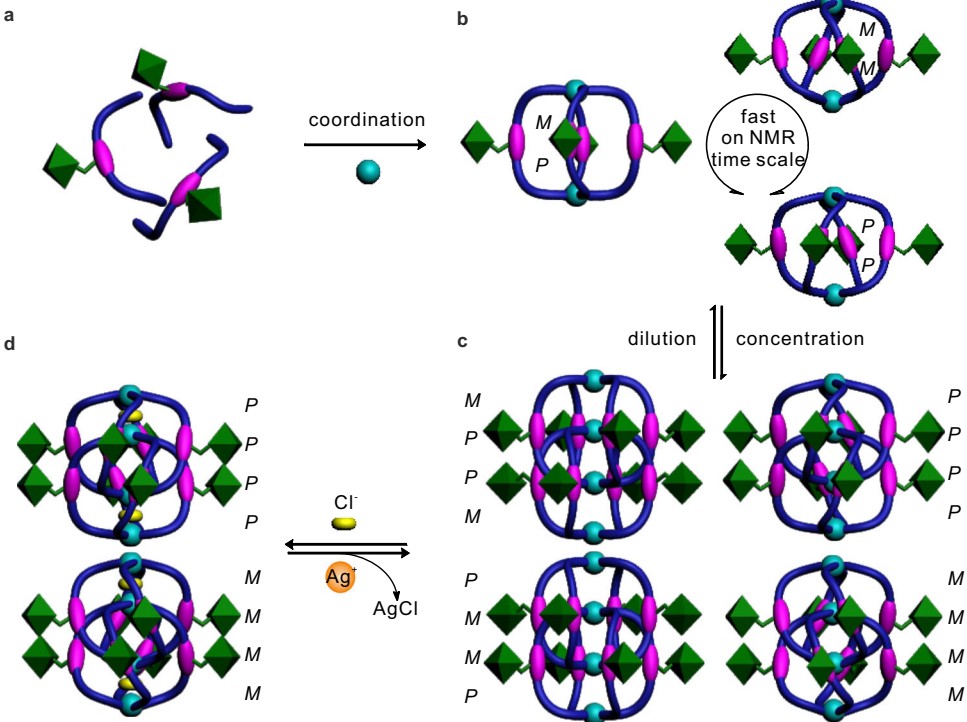

**Fig. 3 Schematic representation of the chirality dynamics and hierarchical communication of the helicates. a** A mixture of sequences with different twisting conformations. **b** The monomeric helicates with three conformations (*PM*, *MM*, and *PP*) in fast exchange on the NMR timescale. **c** Four isomers (*PPPP*, *MMMM*, *PMMP*, and *MPPM*) of the dimeric helicates exist due to the communication of the ligands. **d** Two conformations (*PPPP* and *MMMM*) of the dimeric helicates complexed with two Cl⁻ anions.

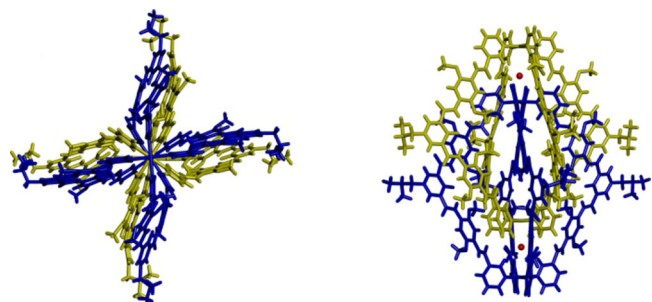

**Fig. 4 Crystal structures of [2Cl⊂2].** Top (left) and side (right) views of the crystal structures of [2Cl⊂2]. Only the *PPPP* enantiomer is shown. The structures belong to the centrosymmetric space groups and thus also contain the *MMMM* enantiomer. The solvent molecules, anions outside of cavity, and disorders have been removed for clarity.

by a larger upfield chemical shifts of the corresponding protons (l and m) relative to the homogenous protons (g and f) on which no stacking occurs (Fig. 2c).

**Control of chirality communication by ion stimulus.** The cavities of **2** offer its capability to encapsulate anions, and we inferred that this encapsulation would further approach the ligands and strengthen the chirality communication. Upon the addition of NBu₄Cl to a mixture of **1** and **2** (ca. [**L**] = 1.8 mM), a concentration at which the dimeric helicate predominates (60%), the NMR signals of **1** and **2** disappeared, and new signals emerged (Fig. 2d, Supplementary Fig. 20). Job's plots derived from NMR titration experiments revealed a 1:2 binding stoichiometry for **2** to chloride anions (Supplementary Fig. 21). We thus attributed this new species to the complex [2Cl⊂2], which

was supported by the ESI mass technique as well (Supplementary Fig. 22). The X-ray crystal structures of [2Cl⊂2] were obtained and confirmed the interlocking and twisting structure, and stoichiometry of the host-guest complex (Fig. 4). All the ligands are disposed in a helical fashion subtending an average azimuthal angle[24] of 30 degree with respect to the Pd···Pd axis (Supplementary Fig. 25), despite some segments twist less regularly due to the crystal packing effects among the adjacent helicates (Supplementary Fig. 26). The two negatively charged chloride anions are evidently located in the outer cavities of the complex, in line with the observation of a significant downfield shift of the helicate's hydrogen atoms (protons q and a) pointed inward. Only one pair of enantiomers (*PPPP* and *MMMM*) was obtained from the crystal structures, which is in agreement with the NMR experiments, showing the number of signals of [2Cl⊂2] reduced by half compared to that of **2** (Fig. 2). Such chirality selectivity apparently results from a preference of the structural change induced by the Cl⁻ filling. Upon the encapsulation of Cl⁻ anions, the interpenetrated helical structure must be shortened by the electrostatic attractions between Cl⁻ and Pd²⁺, with a Pd···Pd (an internal and the nearest outer one) distance of 6.4 Å. As observed from crystal structures, the largest aromatic overlapping area of ligands moved to the segments between the inner pyridine rings of one monomeric helicate and the outer phenyl rings of the other crossed helicate, because of the shortening of the dimeric helicate architecture. Accordingly, the ¹H NMR signals of the corresponding phenyl protons (i.e., protons f and g) underwent an upfield chemical shift. In contrast, the proton signals (l and m) showed a downfield shift a little bit thanks to deshielding (Fig. 2c, d). Further elucidation of the conformations (between *PPPP* and *PMMP*) of imitative dimeric helicates [2Cl⊂2] showed the aromatic stacking of *PPPP* isomer was more effective than that of

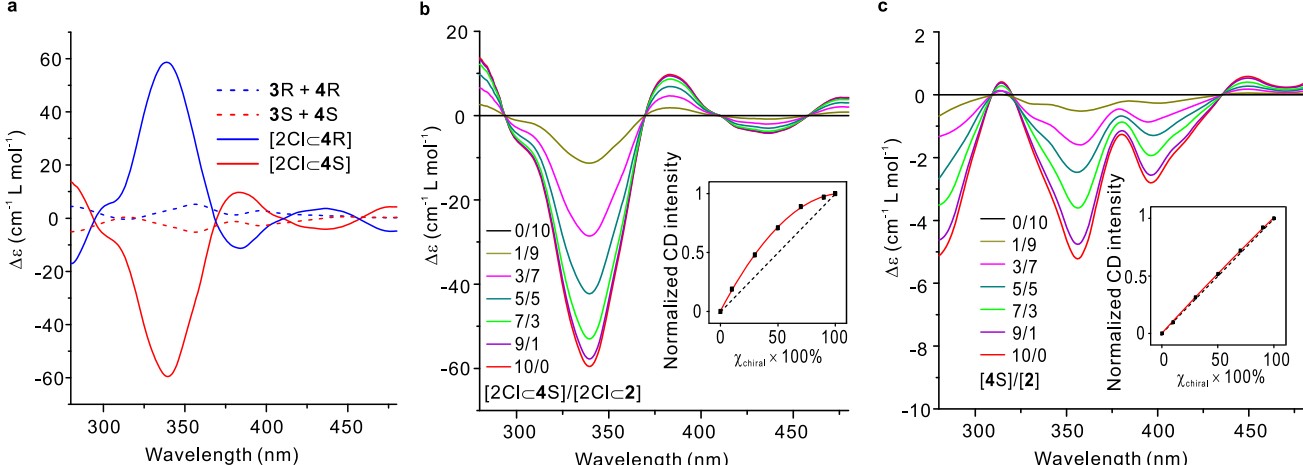

**Fig. 5 Chiral transmission and amplification from side chains to backbones. a** Circular dichroism spectra in CH$_3$CN of **3** R and **4** R mixture ([**3** R] + 2[**4** R] = 1×10$^{-4}$ M, blue dashed line), **3** S and **4** S mixture ([**3** S] + 2[**4** S] = 1×10$^{-4}$ M, red dashed line), [2Cl⊂**4** R] (1×10$^{-4}$ M, blue solid line), and [2Cl⊂**4** S] (1×10$^{-4}$ M, red solid line). **b** Main figure, circular dichroism spectra in CH$_3$CN for mixtures of [2Cl⊂**2**] and [2Cl⊂**4** S] at various molar ratios at equilibria (the total concentration is kept constant at 1×10$^{-4}$ M). Inset, a plot of the normalized intensities at 331 nm versus the percentages of [2Cl⊂**4** S]. **c** Main figure, circular dichroism spectra in CH$_3$CN for mixtures of **2** and **4** S at various molar ratios at equilibria (the total concentration is kept constant at 1×10$^{-4}$ M). Inset, a plot of the normalized intensities at 359 nm versus the percentages of **4** S.

*PMMP* isomer, which may explain the chiral preference (Supplementary Fig. 27).

The [2Cl⊂**2**] complex could be recovered to helicate **2**, which was achieved by the addition of Ag$^+$ to the solution of complex [2Cl⊂**2**], while the Cl$^-$ anions were gradually extracted from the cavities to generate AgCl precipitates. The distance between each of the monomeric helicates extends during this process, and the chirality communication is weakened. The result is supported by the regeneration of signals for the four isomers of the dimeric helicate as shown by $^1$H NMR spectra (Supplementary Fig. 24). Therefore, the chirality of the sequences can be strongly or weakly communicated by responding to ion stimulus in a further step beyond the changing of concentration.

**Regulation of cooperative chirality communication from side chains to backbones.** The performance of chiral communication in all stepwise processes is based on the spatial approach of the sequences. To additionally explore the detailed spatial effect to the information communication in this system, the ligands **L**$^R$ and **L**$^S$ with chiral substituents as probes were prepared. We hypothesized that the chirality communication would happen on the side chains themselves when they were in proximity beyond the communication of the backbones. Furthermore, we envisaged that the chirality information of the side chains could be transferred to the backbones, and thus might lead to homochirality of all the ligands after the final control step. It is of particular interest to obtain the homochiral architecture, as it could get the maximum of chiral signals. The $^1$H NMR spectra (Fig. 2e) of the interpenetrated helicate with enantiomeric chiral side chains showed an octuplet splitting of signals, indicating the existence of four nearly equal diastereoisomers (for instance, the four diastereoisomers of **4** S were attributed to [**PPPP**]$^S$, [**MMMM**]$^S$, [**PMMP**]$^S$, and [**MPPM**]$^S$). Consistent with **2**, the interpenetrated helicates of **4** S could also encapsulate two Cl$^-$ anions. The $^1$H NMR (Fig. 2f) show a new species assigned to complex [2Cl⊂**4** S] emerges upon the addition of NBu$_4$Cl to a solution of **4** S in CD$_3$CN. The number of NMR signals observed for [2Cl⊂**4** S] reduces by a quarter compared to that of **4** S, suggesting that the [2Cl⊂**4** S] complex is formed with only one conformer, which was speculated to be [**PPPP**]$^S$ or [**MMMM**]$^S$ alternatively, based on the similar complexation of **2** to [2Cl⊂**2**]. The chirality

communications of the sequences **L**$^R$ and **L**$^S$ can also be finely monitored by circular dichroism (CD) measurements thanks to the introduction of chiral substituents. Of note is the fact that the chiral centers of the ligands were not close enough to the first coordination sphere to prevent from having a strong bias for helicity. Therefore, the helicates **3** R and **3** S, together with their dimer forms **4** R and **4** S show a mirror image CD behavior with relatively weak absorptions at 250–450 nm (Fig. 5a). The impotent CD is likely due to the diastereo-isomerization of the helicate by the chiral side chains rather than their induction to the helicate scaffold. In striking contrast, the [2Cl⊂**4** R] and [2Cl⊂**4** S] complexes show clear CD absorption at around 340 nm, which could be attributed to the preferential twisting of the framework induced by the chiral side chains. These results again support that the structure of interpenetrated helicate can be shortened and contracted by encapsulation of Cl$^-$ anions, whereby the corresponding proximity between the chiral side chains as well as between them and helical skeleton is close enough to communicate and induce the chirality preference. In this way, the chirality information was accumulated from the chiral side chains and transferred to the helicate skeleton, which unified the helicity and enabled one out of ten diastereoisomers to be selected.

The chirality correlations in our model also display a cooperative behavior with a large chiral induction resulting from the addition of a small amount of chiral complexes to achiral complexes, i.e., the addition of [2Cl⊂**4** S] to [2Cl⊂**2**]. Mixing [2Cl⊂**2**] and [2Cl⊂**4** S] at equilibrium gave rise to a purely statistical mixture of the heteromeric complex through the ligand exchange process, which was reflected by the appearance of broadening of multiple signals in the $^1$H NMR spectra (Supplementary Fig. 38). CD spectra of these heteromeric complexes were monitored with variable proportions of [2Cl⊂**2**] and [2Cl⊂**4** S] while maintaining a total concentration equal to 1×10$^{-4}$ M. As shown in Fig. 5b, the CD intensities at 331 nm increased with the rise in the proportion of the chiral helicate [2Cl⊂**4** S], and this increase deviated positively from linearity. A control experiment (Supplementary Fig. 39) was carried out between **2** and **4** S, the side chains of those mixtures we assumed were not close to each other. By contrast, the experiment reveals a linear CD signal change as no side chain communication occurred in this situation (Fig. 5c).

## Discussion

We present an example of how aromatic oligoamide sequences can communicate helicity in a hierarchical manner, a feature, which allows the communication to be implemented in multiply reversible controls and the chirality information to display at different levels. The chirality of the sequence is originated from the twisting nature of the amide segments, and the communication and selectivity of its helical chirality is dependent on the sterically crowded environment. The steric restriction steps for chiral control are shown in Fig. 3: (i) coordination of the sequences to the monomeric helicate; (ii) dimerization of the monomeric helicates into the dimeric form; (iii) encapsulation of $Cl^-$ anions leading the dimeric helicate to a contraction state. Besides the interactions of the helical skeletons, the information regarding to handedness could be synergistically converged from the chiral side chains and transferred to the ligand backbones with amplification of the helical chirality. This cooperative behavior from one kind of chiral communication (chirality of side chain) response to another (helical chirality) is relative to an especial approach to the classical Sergeants and Soldiers effect[5].

The reversibly hierarchical regulation of chirality communication property as described here may be of use in designing stimuli-responsive chiral sensors[47,48] and pave the way to the molecular machines[49] for modelling of information communication and processing.

## Methods

**General procedure for self-assembly of the monomeric and dimeric helicates**. Take the case of the monomeric helicate **1** and dimeric helicate **2**, to a solution of $CD_3CN$ (500 μL) in a NMR tube were added the ligand **L** (0.67 mg, 0.90 μmol) and $Pd(CH_3CN)_4(BF_4)_2$ (0.20 mg, 0.45 μmol). The reaction mixture was heated at 56 °C for 12 h to give a mixture solution of the helicates (the ratio of **1** and **2** can be adjusted by decreasing or increasing the concentration). The solution was used for further NMR studies. In a similar way, the helicates **3** and **4** were assembled by mixing the corresponding chiral ligands with $Pd(CH_3CN)_4(BF_4)_2$.

**NMR studies of capture and release of chloride anions to trigger the chirality**. In a NMR tube, the above monomeric and dimeric helicates were heated with $NBu_4Cl$ (0.45 μmol, 25.7 μL of a 17.5 mM stock solution in $CD_3CN$) at 56 °C for 12 h to give a 0.21 mM solution of the $Cl^-$ complexes. Upon addition of $AgBF_4$ (1.35 μmol), the $Cl^-$ complexes would be returned to the original state of the helicates by incubating for around 15 days at room temperature, monitored by $^1H$ NMR.

**Circular dichroism**. The equilibrations of helicate samples were performed first and monitored by $^1H$ NMR spectroscopy, and CD spectra were recorded immediately after dilution the samples with $CH_3CN$.

## Data availability

Crystallographic data and experimental details of the structural refinement for X-ray crystal structure of [2Cl⊂**2**] have been deposited at the Cambridge Crystallographic Data Centre under deposited no. CCDC 1893706. The data can be obtained free of charge from the Cambridge Crystallographic Data Centre (http://www.ccdc.cam.ac.uk/data_request/cif). All other data that support the findings of this study are available within the paper and its Supplementary Information files or from the corresponding author upon reasonable request.

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

## Acknowledgements

This work was supported by the National Natural Science Foundation of China (no. 21871101). Special thanks are also given to the Analysis and Testing Center in Huazhong University of Science and Technology for their help with material characterizations.

## Author contributions

J.Z. and Q.G. conceived the project. J.Z. designed, synthesized, and tested the compounds. C.M. performed the X-ray crystallographic analysis. D.L. and L.H. carried out the DFT modelling. Q.G. and J.Z. wrote the paper. All authors discussed the results and commented on the manuscript.

## Competing interests

The authors declare no competing interests.
