## [Peer Review File · Nature Communications]

Reviewers' comments:

Reviewer #1 (Remarks to the Author):

The manuscript by Gan and co-authors describes an observation on chiral transfer of concentration dependent interpenetrated dimerization cages by coordination of pyridine moiety in aromatic oligoamide with Pd, and the chiral transfer can be enhanced by encapsulation of Cl anion subsequently. These results as supported by experimental evidences of nmr, XRD and CD along with energy simulation are very interesting and deserves to be published in Nat. Commun, but the following comments should be addressed before acceptance:

(1) The present discussion is largely based on chemical shift and signal splitting in nmr, but the mechanism of the chiral transfer is not discussed clearly.

(2) Figure 2c & S4, compared with proton r, why protons o, p and q show significant chemical shift with different isomer?

(3) Line 12 at page 4, based on modeling, the top segment of the upper cage and bottom segment of the lower cage are quite flexible, why the isomers of MMMP and PPPM are not considered?

(4) When encapsulation the Cl anion experiments, the formation process should be more clarified. The formation of dimeric cages [Cl₂] and [Cl₄S] is available, is it possible that the bottom Pd of upper cage and top Pd of lower cage capsule one Cl anion? If so, the chiral transfer occurs in this condition or not?

(5) Is there hydrogen bond between protons k,h and central carbonyl oxygen? For better understanding the twist arrangement of amide, I suggest VCD experiment for 3S, 4S and [2Cl₄S].

(6) Compared with [2Cl₂], why proton n of [2Cl₄S] shows obvious chemical shift?

(7) In synthesis section, the empirical formula not correct from compound 7, all are C₃₂H₃₂N₆O₄.

(8) Figure 3b, the left illustration is MP or PM?

Reviewer #2 (Remarks to the Author):

Zhang et al. explored metallohelicite formation by coordination of Pd(II) with aromatic tetraamides terminated with a pyridine motif. While a dimeric form was exclusively present at low concentration, interpenetrated helicates were found at higher concentration as a mixture of two pairs of enantiomers (PPPP/MMMM and PMMP/MPPM). Upon binding of Cl⁻ anion in the cavity, the helicates were further rectified to only allow one enantiomer pair (PPPP/MMMM) presumably due to the stronger pi-pi interaction. The authors also showed that a chiral version of the tetraamide formed the helicates and the chiral information can be amplified in the Cl⁻ complex state.

Compared to the previous studies of metallohelicites, use of the oligoamide scaffold that can twist shed a new insight on the hierarchical process of chirality selection during self-assembly. The authors thoroughly identified the self-assembled structures by combining different techniques, and

their conclusion seems sound. I recommend publication of this manuscript in Nature Communications after addressing following questions/comments:

1. Introduction: the authors emphasized importance of chirality communication in flexible self-assemblies. To me, however, the essence of this work is how the specific chirality can be selected in each stage of the self-assembly process. I would like to suggest to consider revising the Introduction section to provide more an appealing and relevant discussion.
2. Chirality selection process: the authors attributed pi-pi interaction to the key of both chirality selection process for the dimeric helicate formation and also upon the Cl⁻ binding. More detailed and quantitative discussion would be helpful to the readers. Also, do the authors see any difference in chemical shifts in the ¹H NMR spectra which may be correlated to the extent of pi-pi interaction?
3. Figure 3 needs a legend.
4. Observation of chirality amplification only in the Cl⁻ complex state is interesting. The ligand exchange pattern seems different from the interpenetrated helicate state. Do the authors suggest that the ligand exchange will result in a purely statistical mixture of L and L^ΔS, or the mixing will occur between the enantiomeric pairs (i.e., PPPP and PPPP^ΔS)?

Reviewer #3 (Remarks to the Author):

In this paper, the Pd₂L₄ complexes and its dimeric interpenetrated Pd₄L₈ complexes were synthesized from Pd²⁺ ion and aromatic oligoamide bispyridine ligands L. The major claim of the paper is that the control of helical conformation in the complex was achieved by the dimerization induced by concentration change (Fig. 3, b to c) or chloride anion (Fig. 3, c to d), and by the introduction of side chains with stereogenic centers (3R, 4R (or 3S, 4S)). The claim, however, is not fully supported by the experiments in this manuscript. Two main concerns are as follows.

(I) When Fig. 2b (0.2 mM) and Fig. 2c (8.0 mM) are compared, the species in Fig. 2c gave 4 times as many NMR signals as monomer 1. The authors concluded that dimer 2 exists as the equimolar mixture of two conformational diastereomers, that is, [PPPP (or MMMM)] and [PMMP (or MPPM)]. This claim is not inconsistent with the discussion based on symmetry and DFT calculations, and it is still a matter of speculation. According to the authors' interpretation, the two diastereomers of 2 ([PPPP (or MMMM)] and [PMMP (or MPPM)]) have the same thermodynamic stability. In the case of compounds 4S with chiral side chains, the four diastereomers (PPPP, MMMM, PMMP, and MPPM) exist in equimolar amounts. However, it would be rare that the two or four diastereomers have nearly equal stability. The authors should consider a possibility that the product is one species with Pd₄L₈ composition, but its structure is somehow desymmetrized (not in the manner of P/M helical twisting) to exhibit twice the number of signals than expected for the interpenetrated dimer. This possibility cannot be excluded from the reported experiments. The following experiments might help to reveal the phenomenon: (a) 1H-1H NOESY experiments to observe correlations between individual cages (e.g. J. Am. Chem. Soc. 2016, 138, 794-797 use the technique to determine the interlocked structure without ambiguity). (b) Analysis of EXSY peaks in the NOESY spectrum (at an elevated temperature, if necessary) to discuss the chemical exchanges between "diastereomers". (c) X-ray structure of the species in the state of Fig. 3c.

(II) The authors insisted that the crystal structure [2Cl@2] has the PPPP (or MMMM) helicity. Somehow, this is NOT RIGHT. Below I summarize the dihedral angles of pyridine ring and amide group (defined as the torsion angle of the four atoms selected as the attached file). Here, positive values correspond to M helicity according to the definition in Figure 1. Atom numbers correspond to those in the CIF file.

· The corresponding angles of four pyridines coordinating to Pd2 are (44.52°, -31.09°, 44.52°, -31.09°), thus MPMP

· The corresponding angles of four pyridines coordinating to Pd1 are (-5.89°, -19.20°, -5.89°, -19.20°), thus PPPP

· The corresponding angles of four pyridines coordinating to Pd3 are (-28.28°, -9.45°, -28.28°, -9.45°), thus PPPP

· The corresponding angles of four pyridines coordinating to Pd00 are (41.82°, -20.12°, 41.82°, -20.12°), thus MPMP

In other words, it is not appropriate to describe the helicity of [2Cl@2] simply as "PPPP" (or "MMMM").

To add more, in relation to the concern (I), the torsions of the pyridine ring and the amide are generally small. It is questionable whether the activation barrier between the diastereomers of chloride-free 2 (PPPP and MPPM) is large enough to allow the mutual conversion slower than 1H NMR timescale.

Thus, the main story of this paper (Figure 3) is not supported by the experiments. Even if the authors' claim is true, the chemistry is limited to the structural analysis of Pd2L4 cage and its interpenetrated dimeric complex, and the similar compounds have already been reported in previous studies (refs. 23-31). The paper would not appeal to a wide range of scientific readers. The paper is not recommended for publication in Nature Communications.

Additional comments.

· Pd2L4 complexes reported by Yoshizawa et al. should be included as references (e.g. J. Am. Chem. Soc. 2011, 133, 11438–11441). It has a helicity, and its ligand motion is so restricted that the motion can be observed by 1H NMR.

· Investigation of anions other than BF4- and Cl- might be helpful (NO3-, Br-, F-, etc.).

· On page 2, 7 line from the top, "in despite of" should be "in spite of".

· On page 4, 8 line from the top, "might also cause by" should be "might also be caused by".

· On page 7, 5 line from the top, "the close the distance" should be "the closer the distance".

Reviewers' comments:

Reviewer #1:

The manuscript by Gan and co-authors describes an observation on chiral transfer of concentration dependent interpenetrated dimerization cages by coordination of pyridine moiety in aromatic oligoamide with Pd, and the chiral transfer can be enhanced by encapsulation of Cl anion subsequently. These results as supported by experimental evidences of nmr, XRD and CD along with energy simulation are very interesting and deserves to be published in Nat. Commun, but the following comments should be addressed before acceptance:

(1) The present discussion is largely based on chemical shift and signal splitting in nmr, but the mechanism of the chiral transfer is not discussed clearly.

Response: We thank this reviewer for these valuable comments. The chirality of the sequence is originated from the twisting nature of the amide moiety, and the communication and selectivity of its helical chirality is dependent on the restriction of the steric crowding of the sequence. The steps for chiral control through manipulation of steric crowding are shown in Figure 3: (i) coordination of the sequences to the monomeric helicate; (ii) dimerization of the monomeric helicates into the dimeric form; (iii) encapsulation of Cl⁻ anions leading the dimeric helicate to a contraction state. These comments are now presented in detail in the discussion sections.

(2) Figure 2c & S4, compared with proton r, why protons o, p and q show significant chemical shift with different isomer?

Response: In the modelling structures (see Figure S23) of interpenetrated helicate **2**, the protons r at 2-position of the inside pyridine are pointed to the inner cavity and are shielded by the methoxyl (-OCH₃) group from the adjacent ligands (evidenced by NOE cross peaks, see Figure S7). Therefore, its chemical shifts may not change too much for different isomers. By contrast, the protons o, p and q (protons at 4-, 5-, and 6-position of the inner pyridine) are in the “groove” of the helicate and pointed out to the external environment. We infer the different locations of BF₄⁻ counter-ions in different isomers may vary the chemical shifts of these protons.

Figure. The structure of dimeric helicate with protons assignment.

(3) Line 12 at page 4, based on modeling, the top segment of the upper cage and bottom segment of the lower cage are quite flexible, why the isomers of MMMP and PPPM are not considered?

Response: In the structure of the interpenetrated helicate, all the ligands are mechanically interlocked and spatially correlated. The models of all the isomers of **2** show that the energy of PPPM isomer is 8.3 kcal mol⁻¹ larger than that of PPPP isomer. This preference is probably due to the better contacts between the two individual helicites with ‘like’ twisting conformations (e.g., PP + PP, or PM + MP). From another point, symmetry is generally favorable calling for the supramolecular architectures, and the asymmetric helicites (such as MMMP and PPPM) might not be preferred.

(4) When encapsulation the Cl anion experiments, the formation process should be more clarified. The formation of dimeric cages [ClC2] and [ClC4S] is available, is it possible that the bottom Pd of upper cage and top Pd of lower cage encapsulate one Cl anion? If so, the chiral transfer occurs in this condition or not?

Response: We thank this suggestion. To figure out how many chloride anions can be located into the three cavities, we did Job’s plots from NMR titration experiments. It confirms a 1:2 binding stoichiometry for the dimeric helicate to chloride anions. Moreover, the proton r which pointed into the central cavity did not shift much before or after the addition of chloride anions. Crystal structures also support that two chloride anions encapsulated into the outer cavities. However, we can not rule out that the chloride anion may also fast enter in and out of the central cavity, and extremely weakly interacted with the dimeric helicate. Whether or not this happens, the chiral transfer exists, since NMR and CD did not altered when addition more Cl anions after saturation of the complex. Addition over 5 equiv. of Cl anions leads to precipitation of the complex.

Figure S26. Job's plot of helicite with $n\text{BuN}^+\text{Cl}^-$. Maintain a total concentration equal to $[\mathbf{1}]/2 + [\mathbf{2}] + [\text{Cl}^-] = 0.5 \text{ mM}$, and define the ratio of $\text{Cl}^-/\text{total concentration}$ as a molar fraction $\chi = [\text{Cl}^-]/([\mathbf{1}]/2 + [\mathbf{2}] + [\text{Cl}^-])$. The inflexion point at 0.66 shows a 1:2 (helicite/ Cl^-) stoichiometry. The plot is missing at the range of high ratio of Cl^- due to the precipitation of the complex.

(5) Is there hydrogen bond between protons k,h and central carbonyl oxygen? For better understanding the twist arrangement of amide, I suggest VCD experiment for 3S, 4S and [2Cl⊂4S].

Response: We thank this reviewer for the crucial advice. Exam of the vibrational state is a good experimental approach to test the twist arrangement of amide group. However, attempts to assess the vibrations by vibrational circular dichroism (VCD) failed due to the poor solubility of the metal complex. Instead, we have tried infrared spectroscopy to evaluate the vibration of amide group. The compounds of the ligand \mathbf{L}^s , monomeric helicite $\mathbf{3S}$, dimeric helicite $\mathbf{4S}$, and $[\mathbf{2Cl}\subset\mathbf{4S}]$ complex show very similar infrared spectra, and there was no significant change at the range of the stretching and bending vibrations of amide ($1800\text{-}1400 \text{ cm}^{-1}$). We speculate this is due to the overlapping of signals as there may be both twisting and untwisting amide moieties in the molecular skeleton. Two absorption bands ($2900\text{-}3100$, $3400\text{-}3600 \text{ cm}^{-1}$) for the stretching vibration of N-H have been observed, thus we infer that the central amide is hydrogen bonded whereas the outer amide does not.

Figure. The infrared spectra of the ligand L^S , monomeric helicate **3S**, dimeric helicate **4S**, and $[2Cl\subset 4S]$ complex.

(6) Compared with $[2Cl\subset 2]$, why proton n of $[2Cl\subset 4S]$ shows obvious chemical shift?

Response: The peaks n/e are referred to the outer amide protons of the sequences. As shown in the following spectra, by comparing chiral dimeric helicate $[2Cl\subset 4S]$ with non-chiral helicate $[2Cl\subset 2]$, it is true that these protons show more chemical shifts than other protons. And this phenomenon is also true for monomeric helicates and even for just ligands. Apparently, it is the difference of side chains that results in the variation of chemical shift of protons n/e . We conjecture that the different size of the side chains allows the sequences to suffer solvation differently, and the active nature of the amide protons (n/e) make them more susceptible to solvation.

Figure. 1H NMR spectrum (400 MHz, 298 K) of ligand **L**, L^S , monomeric cage **1** and **3S**, $[2Cl\subset 2]$ and $[2Cl\subset 4S]$.

(7) In synthesis section, the empirical formula not correct from compound 7, all are

C32H32FN6O4.

Response: We are sorry for our incorrect writing and carelessness. And we have checked and corrected the following mistakes in the synthesis section. Compound **7**, C₁₄H₁₅N₃O₃; Compound **9**, C₁₁H₁₃NO₅; Ligand **L**, C₃₉H₃₉N₇O₉; Compound **11**, C₁₄H₁₉NO₇; Compound **12**, C₁₇H₂₃NO₇; Compound **13**, C₁₃H₁₅NO₇; Ligand **L^R** (or **L^S**), C₄₁H₄₁N₇O₁₁.

(8) Figure 3b, the left illustration is MP or PM?

Response: We apologize for the confusion. We have modified the figure 3b, hopefully it is easier to identify the structural conformation now.

Reviewer #2:

Zhang et al. explored metallohelicite formation by coordination of Pd(II) with aromatic tetraamides terminated with a pyridine motif. While a dimeric form was exclusively present at low concentration, interpenetrated helicates were found at higher concentration as a mixture of two pairs of enantiomers (PPPP/MMMM and PMMP/MPPM). Upon binding of Cl⁻ anion in the cavity, the helicates were further rectified to only allow one enantiomer pair (PPPP/MMMM) presumably due to the stronger pi-pi interaction. The authors also showed that a chiral version of the tetraamide formed the helicates and the chiral information can be amplified in the Cl⁻ complex state.

Compared to the previous studies of metallohelicites, use of the oligoamide scaffold that can twist shed a new insight on the hierarchical process of chirality selection during self-assembly. The authors thoroughly identified the self-assembled structures by combining different techniques, and their conclusion seems sound. I recommend publication of this manuscript in Nature Communications after addressing following questions/comments:

1. Introduction: the authors emphasized importance of chirality communication in flexible self-assemblies. To me, however, the essence of this work is how the specific chirality can be selected in each stage of the self-assembly process. I would like to suggest to consider revising the Introduction section to provide more an appealing and relevant discussion.

Response: We thank this reviewer for these supportive comments and crucial advice. The discussions about the chirality selection at each step are now presented in detail in the introduction section. "In the present study, we show that the chirality of the aromatic oligoamide sequences derived from amide twisting can be progressively communicated and regulated in a hierarchical manner through steric crowding manipulation. Specifically, the amide twisting orientation can be firstly correlated and unified by coordination of four sequences into a M₂L₄ helicate structure. Mechanically interlocking of two single helicates into a dimeric helicate via concentration further makes the ligands close to each other, leading to the restriction of the conformational freedom of the amide twisting and the selection of chirality. The intervals between the ligands in the interlocked structure can be even shortened by encapsulation of Cl⁻ anions, ultimately making the flexibility of

twisting frozen and a homochirality of the ligands.”

2. Chirality selection process: the authors attributed pi-pi interaction to the key of both chirality selection process for the dimeric helicate formation and also upon the Cl⁻ binding. More detailed and quantitative discussion would be helpful to the readers. Also, do the authors see any difference in chemical shifts in the ¹H NMR spectra which may be correlated to the extent of pi-pi interaction?

Response: We thank this reviewer for this suggestion. The selection of helical chirality was achieved by hierarchical shortening of the intervals between the sequences. Dimerization of monomeric helicate into dimeric helicate is one way to gather the ligands more closely, and we think this dimerization process is driven by π - π stacking. In turn, to gain the maximum of π - π interaction it requires the ligands with the same twisting at the intertwined section, resulting in the chirality selection of the dimeric helicates (PPPP/MMMM and PMMP/MPPM). In the modelling structure of dimeric helicate, a large overlapping area of ligands happens between the phenyl rings at the intertwined section, and it is supported by a larger upfield chemical shifts of the corresponding protons (l and m) relative to the homogenous phenyl protons (g and f) for which no stacking occurs. This mechanism of the chirality selection also works well on the Cl⁻ complex. The interpenetrated helicate structure must be shortened by the electrostatic attractions between Cl⁻ and Pd²⁺, resulting in more close proximity between ligands and thus a higher chirality selection. The incorporation of the negatively charged guests inside **2** led to a significant downfield shift of the helicate’s hydrogen atoms (protons q and a) that are pointing inside the chloride-filled cavities. Meanwhile, as observed from crystal structures, the most overlapping area of ligands moved to the segments between the inner pyridine rings of one monomeric helicate and the outer phenyl rings of the other crossed monomeric helicate, because of the shortening of the dimeric helicate structure. Accordingly, the ¹H NMR signals of the corresponding phenyl protons (i.e., protons f and g) underwent an upfield chemical shift. In contrast, the proton signals (l and m) shown a downfield shift a little bit thanks to deshielding. We have commented these to the main text.

Figure. a) ¹H NMR spectra of the helicate **1** and **2**, and the corresponding Cl⁻ complex. b) Schematic representation of helicate structures showing the overlapping area.

3. Figure 3 needs a legend.

Response: Thank you for pointing this. We have modified and added a new legend for Figure 3.

“Figure 3. Schematic representation of the chirality dynamics and hierarchical communication of the helicates. a, A mixture of sequences with different twisting conformations. **b,** The monomeric helicates with three conformations (*PM*, *MM*, and *PP*) in fast exchange on the NMR time scale. **c,** Four isomers (*PPPP*, *MMMM*, *PMMP*, and *MPPM*) of the dimeric helicates exist due to the communication of the ligands. **d,** Two conformations (*PPPP* and *MMMM*) of the dimeric helicates complexed with two Cl⁻ anions.”

4. Observation of chirality amplification only in the Cl⁻ complex state is interesting. The ligand exchange pattern seems different from the interpenetrated helicate state. Do the authors suggest that the ligand exchange will result in a purely statistical mixture of L and L[^]S, or the mixing will occur between the enantiomeric pairs (i.e., *PPPP* and *PPPP*[^]S)?

Response: Since the ligand exchange is uncontrollable in this system, we believe that it will produce a purely statistical mixture in the absence of any bias. The exchange NMR experiments of mixing chiral and achiral helicates at equilibrium displayed a broadening of NMR signals, which is far from the situation that the helicates were chiral self-sorted (i.e., *PPPP* and *PPPP*[^]S). See Figures S34 and S35 of the supporting information. We added a more clear description in the text.

Reviewer #3:

In this paper, the Pd₂L₄ complexes and its dimeric interpenetrated Pd₄L₈ complexes were synthesized from Pd²⁺ ion and aromatic oligoamide bispyridine ligands L. The major claim of the paper is that the control of helical conformation in the complex was achieved by the dimerization induced by concentration change (Fig. 3, b to c) or chloride anion (Fig. 3, c to d), and by the introduction of side chains with stereogenic centers (3R, 4R (or 3S, 4S)). The claim, however, is not fully supported by the experiments in this manuscript. Two main concerns are as follows.

(I) When Fig. 2b (0.2 mM) and Fig. 2c (8.0 mM) are compared, the species in Fig. 2c gave 4 times as many NMR signals as monomer 1. The authors concluded that dimer 2 exists as the equimolar mixture of two conformational diastereomers, that is, [PPPP (or MMMM)] and [PMMP (or MPPM)]. This claim is not inconsistent with the discussion based on symmetry and DFT calculations, and it is still a matter of speculation. According to the authors' interpretation, the two diastereomers of 2 ([PPPP (or MMMM)] and [PMMP (or MPPM)]) have the same thermodynamic stability. In the case of compounds 4S with chiral side chains, the four diastereomers (PPPP, MMMM, PMMP, and MPPM) exist in equimolar amounts. However, it would be rare that the two or four diastereomers have nearly equal stability. The authors should consider a possibility that the product is one species with Pd₄L₈ composition, but its structure is somehow desymmetrized (not in the manner of P/M helical twisting) to

exhibit twice the number of signals than expected for the interpenetrated dimer. This possibility cannot be excluded from the reported experiments. The following experiments might help to reveal the phenomenon: (a) 1H-1H NOESY experiments to observe correlations between individual cages (e.g. *J. Am. Chem. Soc.* 2016, 138, 794-797 use the technique to determine the interlocked structure without ambiguity). (b) Analysis of EXSY peaks in the NOESY spectrum (at an elevated temperature, if necessary) to discuss the chemical exchanges between "diastereomers". (c) X-ray structure of the species in the state of Fig. 3c.

Response: We thank this reviewer for the crucial question. The reviewer agrees that the compound **2** is a Pd₄L₈ composition supported by mass spectrum, and she/he supposed another possibility that the splitting of the NMR signals is due to the structural dissymmetry (not in the chirality manner of P/M helical twisting). However, this speculation is conflicted with some results from our system. The Pd₄L₈ compound **4** with chiral ligands show twice as many NMR signals as the compound **2** with achiral ligands. This diastereomeric increase indicates there is already an element of chirality in the structure of compound **2** (i.e., P/M helical element). Otherwise it will give the same number signals upon introducing single stereocenter.

Following the reviewer's suggestion, we have done NOESY experiments to further clarify the structure of Pd₄L₈ compounds. The NOE cross peaks corresponding to the signals of methoxyl (-OCH₃) and pyridine protons (r) were observed (Figure S7): correlations between these protons are unlikely to be intramolecular and are not observed for the single ligand or for monomeric helicate **1**, but may emerge from an intermolecular stacking of the ligands with a slippage motif, as supposed in the interpenetrated structure. The NOE discussions have been added to the main text to more clearly support the interpenetrated structure. The interpenetrated structure of Pd₄L₈ was further supported by NMR chemical shift studies, as we discussed for answering the question 2 of the second reviewer. We did not see any EXSY peaks presumably due to quite slow exchange rate between the diastereomers. Crystals of **2** were also obtained, but their structure could not be solved due to the poor diffraction intensity after many trials. DFT modellings were carried out instead, and the calculation results are consistent with the solution studies.

Figure. ¹H-¹H NOESY spectra (CD₃CN, 400 MHz, 298 K) of **2**.

(II) The authors insisted that the crystal structure [2Cl@2] has the PPPP (or MMMM) helicity. Somehow, this is NOT RIGHT. Below I summarize the dihedral angles of pyridine ring and amide group (defined as the torsion angle of the four atoms selected as the attached file). Here, positive values correspond to M helicity according to the definition in Figure 1. Atom numbers correspond to those in the CIF file.

· The corresponding angles of four pyridines coordinating to Pd2 are (44.52°, -31.09°, 44.52°, -31.09°), thus MPMP

· The corresponding angles of four pyridines coordinating to Pd1 are (-5.89°, -19.20°, -5.89°, -19.20°), thus PPPP

· The corresponding angles of four pyridines coordinating to Pd3 are (-28.28°, -9.45°, -28.28°, -9.45°), thus PPPP

· The corresponding angles of four pyridines coordinating to Pd00 are (41.82°, -20.12°, 41.82°, -20.12°), thus MPMP

In other words, it is not appropriate to describe the helicity of [2Cl@2] simply as "PPPP" (or "MMMM").

To add more, in relation to the concern (I), the torsions of the pyridine ring and the amide are generally small. It is questionable whether the activation barrier between the diastereomers of chloride-free 2 (PPPP and MPPM) is large enough to allow the mutual conversion slower than 1H NMR timescale.

Response: We thank the reviewer for pointing this out. In fact, the azimuthal angle θ is commonly used to describe the helicity for M₂L₄ helicates (e.g. Dalton Trans., 2012, 41, 11273-11275, Angew. Chem. Int. Ed. 2008, 47, 706-710). A structure is qualified to be a helicate with helicity when the azimuthal angle θ is non zero. This method is based on the torsion angle of coordinating segments with respect to the Pd-Pd axis (i.e., N_{pyridine}-Pd-Pd-N_{pyridine} angle), not just in view of the amide torsion angle. In our crystal structure of [2Cl@2], the azimuthal angles for each ligands were measured and summarized in the following table. The azimuthal angles show all the positive charge (we here defined the angle phase as positive if it is anticlockwise spin from the top-to-down view), indicating all the ligands have a coincident helicity (i.e., P helicity).

Figure. Schematic diagrams for the definition of azimuthal angle θ of monomeric and dimeric helicates. Only one ligand per single cage is depicted for clarity. a) *MM* monomeric helicate with a definite azimuthal angle ($\theta < 0$), b) *PP* monomeric helicate ($\theta > 0$), c) the dimeric helicate $[2Cl@2]$ with *PPPP* conformation.

Table. The azimuthal angle θ of the dimeric helicate $[2Cl@2]$ from the crystal structure.

θ	ligand 1	ligand 2	ligand 3	ligand 4
upper helicate	27.61	32.53	27.61	32.53
upper helicate	29.83	27.85	29.83	27.85

Although it is convenient to use this parameter to describe the helicity for M_2L_4 helicates, it has some drawbacks. For example (as we published in *Chem. Commun.*, 2018, 54, 13447-13450), the ligands can be subjected to a distortion or twisting in the center and partitioned into two sections, each of which can thus coordinate the metal ion to different chiralities (*P* and *M*), leading to a meso complex. In this case, the azimuthal angle is equal to zero, but the ligands still show the helical conformation.

Figure. Schematic diagrams for the *MP* monomeric helicate with the azimuthal angle ($\theta = 0$).

In order to better assign the helicity in our system, it is necessary to horizontally divide a monomeric helicate into two equal parts and quantify the helicity in a piecewise manner. Therefore, to characterize the helicity, it needs two symbols (XX , $X = P$ or M) for a monomeric helicate, and four symbols ($XXXX$, $X = P$ or M) for a dimeric helicate, given the consistency of twisting of ligands in the individual helicate. The approach to assign the helicity is presented as follow:

- 1) Determine the azimuthal angle.
- 2) If the azimuthal angle is not zero, observe the angle phase. When it is clockwise spin from the top-to-down view, the corresponding segment is defined with M helicity, and assigned to P helicity when angle phase is anticlockwise.
- 3) If the azimuthal angle is zero, observe the rotation direction of the twisted ligand around the metal. When it is clockwise from the top-to-down view, the corresponding segment is defined with M helicity, and assigned to P helicity when the direction is anticlockwise.

We have added these discussions to the supporting information. This method for P/M helicity definition is consistent with the description in the text where helicity is defined by the helical orientation of each Pd^{2+} cation with respect to the center of the ligand.

We admit some segments show less-regular twisting in the solid state, which was attributed to the crystal packing effects among the adjacent helicates (Supplementary Fig. S37). We also agree that the activation energy for amide twisting should be not large for the ligand and even for the monomeric helicate. This was supported by experiments that only one set of NMR resonances was observed for the ligand and the monomeric helicate, even at low temperature (Supplementary Fig. S3), inferring their isomers are fast exchange in NMR timescale. In contrast, compound **2** show the slow exchange rate between the diastereomers, suggesting a larger activation barrier for the amide twisting. This observation may be interpreted to result from a cooperative behavior during the isomer conversion. The mechanically interlocking of **2** makes the eight ligands spatially correlated, allowing the architecture as a whole to impede the amide twisting with high activation energy barrier. These discussions are now presented in the manuscript.

Thus, the main story of this paper (Figure 3) is not supported by the experiments. Even if the authors' claim is true, the chemistry is limited to the structural analysis of Pd₂L₄ cage and its interpenetrated dimeric complex, and the similar compounds have already been reported in previous studies (refs. 23-31). The paper would not appeal to a wide range of scientific readers. The paper is not recommended for publication in Nature Communications.

Response: keeping with the initial data (splitting signals of NMR, Mass, X-ray, CD), and with additional experiments and discussions revised (such as analysis of chemical shifts of ^1H NMR, and NOESY), we believe that the story of amide twisting behaviors in our systems and their hierarchical control is sufficiently evidenced.

Our title, introduction, main text body and conclusion all emphasize that our novelty claim concerns the discovery and control of a molecular chirality communication behavior in a hierarchical manner. We do not deny that the general structures of interpenetrated helicates have been published before, but we did not claim any novelty on this ground. There are many publications about chirality control, but control in a hierarchical manner is of utmost importance as it gives rise to programming of the

chirality information by responding to different stimuli, which may not be achieved otherwise. We thus believe that a hierarchical control of molecular chirality is a discovery worthy of attention and would appreciate that this claim be assessed for what it is.

Additional comments.

- *Pd2L4 complexes reported by Yoshizawa et al. should be included as references (e.g. J. Am. Chem. Soc. 2011, 133, 11438–11441). It has a helicity, and its ligand motion is so restricted that the motion can be observed by 1H NMR.*
- *Investigation of anions other than BF4- and Cl- might be helpful (NO3-, Br-, F-, etc.).*

Response: Thank you for pointing this, we are sorry for our negligence of the important reference. In this paper, the ligand they used show the equimolar amounts of three atropisomers caused by the hindrance bond rotation. This example is somehow similar to our system that the twisting restriction leads the diastereomers to have nearly equal stability. This reference is cited now. [25] Kishi, N., Li, Z., Yoza, K., Akita, M., Yoshizawa, M. An M₂L₄ molecular capsule with an anthracene shell: encapsulation of large guests up to 1 nm. *J. Am. Chem. Soc.* **133**, 11438–11441 (2011).

The recognition ability of other anions has also been studied by using NMR titration, such as Br⁻, F⁻, I⁻ and NO₃⁻. Nitrate anion can also be filled into the helicate cavity, while others showed very weak binding and made precipitate of the complex. None of these anions can do the reversible job (i.e., trapping and releasing from the helicate). For this reason, chloride anion has been selected and explored in our study for chirality control. While an extensive study of the anion binding behavior of helicate would be an interesting topic for more detailed follow-up work.

Figure. ¹H NMR spectra of the free helicate with addition of different anions.

- *On page 2, 7 line from the top, "in despite of" should be "in spite of".*
- *On page 4, 8 line from the top, "might also cause by" should be "might also be caused by".*
- *On page 7, 5 line from the top, "the close the distance" should be "the closer the distance".*

Response: We corrected these mistakes in the manuscript.

Reviewers' comments:

Reviewer #1 (Remarks to the Author):

The revised manuscript addresses my comments properly, and I agree the publication of this work in Nature Communications.

Reviewer #2 (Remarks to the Author):

The authors revised their manuscript in response to my comments and the quality of the manuscript was improved. I recommend publication of this manuscript in Nature Communications.

Reviewer #3 (Remarks to the Author):

In the first review round, the characterization and interpretation of species in Fig. 2c was questioned. The authors now provide the NOESY spectrum as a new experimental data to support their claim (= the species in Fig. 2 are the equimolar mixture of two conformational diastereomers of the interlocked dimer 2, [PPPP (or MMMM)] and [PMMP (or MPPM)]). The NOE correlation was observed between the methoxy proton and the pyridyl proton r , which itself does not contradict to the authors' claim. Unfortunately, this data is not enough to resolve the concern of the reviewer 3, that is, the product might be a single species with Pd4L8 composition, but its structure is somehow desymmetrized (not in the manner of P/M helical twisting) to exhibit twice the number of signals than expected for the interpenetrated dimer. The author mentioned in the rebuttal letter "The Pd4L8 compound 4 with chiral ligands shows twice as many NMR signals as the compound 2 with achiral ligands. This diastereomeric increase indicates there is already an element of chirality in the structure of compound 2 (i.e., P/M helical element)." However, the chiral nature is not conclusive evidence to show that the observed signal splitting is due to the P/M helical conformer. Any chiral structure would show the signal splitting in use of a chiral ligand.

So far, no experimental evidence has been given to show that the P/M helical twisting is slower than 1H NMR time scale. For the monomeric Pd2L4, the activation energy of the "amide twisting" is so small that the H NMR signals' splittings are not observed at all even at low temperature, 243 K (from this observation, $E_a < 10$ kcal/mol can be estimated) (Fig. S3). According to the authors' claim, the activation energy of the "amide twisting" of the dimeric Pd4L8 becomes so large that the conformational diastereomers are not convertible to each other during the mixing time of NOESY at 298 K (from this observation, $E_a > 18$ kcal/mol can be estimated) (Fig. S7). However, the reasoning for the phenomenon is not clearly presented in the manuscript. The description at lines 32–34, "Mechanically interlocking of two single helicates into a dimeric helicate via concentration further makes the ligands close to each other, leading to the restriction of the conformational freedom of the amide twisting and the selection of chirality." is an abstract and qualitative statement. It does not mention what kind of motion is restricted (rotation around specific bonds, steric hindrance between specific groups, etc.). Without the structural discussion, the readers cannot understand what is happening upon the addition of Cl⁻ anion.

It is strange that the chloride-free 4S exists as the equimolar mixture of four diastereomers (PPPP, MMMM, PMMP, and MPPM) (Fig. 2e), although their structures are so crowded that the P/M conversion cannot be observed by means of NMR. Why is the chirality preference not observed at all from 1H NMR integral ratio at this stage?

As the authors admit, the structure of chloride-bound Pd₄L₈ determined by XRD is not largely twisted (page 11 in the rebuttal letter). Then why does the Cl⁻ shift the equilibrium toward one helical isomer? What is the structural mechanism? Figure 5d is too abstract.

The resubmitted manuscript does not change substantially from the initial one. The main story of this paper (Fig. 3) is not supported by the experiments. The evaluation remains unchanged, and the paper is not recommended for publication in Nature Communications.

Reviewers' comments:

Reviewer #1:

The revised manuscript addresses my comments properly, and I agree the publication of this work in Nature Communications.

Response: We are very grateful to the reviewer for his recognition of our research and for his valuable comments.

Reviewer #2:

The authors revised their manuscript in response to my comments and the quality of the manuscript was improved. I recommend publication of this manuscript in Nature Communications.

Response: We are very grateful to the reviewer for his recognition of our research and for his valuable comments.

Reviewer #3:

In the first review round, the characterization and interpretation of species in Fig. 2c was questioned. The authors now provide the NOESY spectrum as a new experimental data to support their claim (= the species in Fig. 2 are the equimolar mixture of two conformational diastereomers of the interlocked dimer 2, [PPPP (or MMMM)] and [PMMP (or MPPM)]). The NOE correlation was observed between the methoxy proton and the pyridyl proton r, which itself does not contradict to the authors' claim. Unfortunately, this data is not enough to resolve the concern of the reviewer 3, that is, the product might be a single species with Pd4L8 composition, but its structure is somehow desymmetrized (not in the manner of P/M helical twisting) to exhibit twice the number of signals than expected for the interpenetrated dimer. The author mentioned in the rebuttal letter "The Pd4L8 compound 4 with chiral ligands shows twice as many NMR signals as the compound 2 with achiral ligands. This diastereomeric increase indicates there is already an element of chirality in the structure of compound 2 (i.e., P/M helical element)." However, the chiral nature is not conclusive evidence to show that the observed signal splitting is due to the P/M helical conformer. Any chiral structure would show the signal splitting in use of a chiral ligand.

Response: We thank the reviewer for pointing this out. In order to clarify this issue, we have carried out a number of experiments, and found that the monomeric helicate 1-Pt assembled by more kinetically inert metal Pt^{2+} showed two sets of signals, as shown in Figure S12. In addition, the ratio of these two sets of peaks can be biased and regulated by adding different anion guests. This phenomenon cannot attribute to desymmetry of the complex structure on itself, but undoubtedly indicates that there are two isomers on slower exchange in NMR timescale for the monomeric helicate. Taking into account the facts that the amide segments in aromatic amide sequences prefer to distort and deviate the neighboring aromatic rings from coplanarity, which would give rise to the *P* or *M* helical conformation of the molecular strands, we thus speculated that our helicates based on the aromatic amide ligand may also twist in the similar way and be reasonable for the multiple sets of NMR signals.

Text about the monomeric helicate **1-Pt** for P/M helical twisting have been added to the main text and the supplementary information (Figs S16-S17).

Figure S16. ¹H NMR spectrum (CD₃CN, 400 MHz, 298 K) of **1-Pt** with assignment of signals.

Figure S17. Representative ^1H NMR spectrum (CD_3CN , 400 MHz) of **1-Pt** (0.28 mM): a) monomeric helicate **1-Pt** (the mixture of PP/MM and PM); addition of b) 0.50 equiv. and c) 1.0 equiv. SO_4^{2-} to the solution of **1-Pt**; addition d) 0.50 equiv., e) 1.0 equiv. and f) 2.0 equiv. Cl^- to the solution of **1-Pt**. The signs of X_1^- and X_2^- represent Cl^- or SO_4^{2-} anion.

So far, no experimental evidence has been given to show that the P/M helical twisting is slower than 1H NMR time scale. For the monomeric Pd2L4, the activation energy of the “amide twisting” is so small that the H NMR signals’ splittings are not observed at all even at low temperature, 243 K (from this observation, $E_a < 10$ kcal/mol can be estimated) (Fig. S3). According to the authors’ claim, the activation energy of the “amide twisting” of the dimeric Pd4L8 becomes so large that the conformational diastereomers are not convertible to each other during the mixing time of NOESY at 298 K (from this observation, $E_a > 18$ kcal/mol can be estimated) (Fig. S7). However, the reasoning for the phenomenon is not clearly presented in the manuscript. The description at lines 32–34, “Mechanically interlocking of two single helicates into a dimeric helicate via concentration further makes the ligands close to each other, leading to the restriction of the conformational freedom of the amide twisting and the selection of chirality.” Is an abstract and qualitative statement. It does not mention what kind of motion is restricted (rotation around specific bonds, steric hindrance between specific groups, etc.). Without the structural discussion, the readers cannot understand what is happening upon the addition of Cl^- anion.

Response: We thank the reviewer for pointing this out. The P/M helical interconversion mechanism of the aromatic amide sequences is known to take place either via an untwisted or a partially unfolded conformation, both of which need the sequence to undergo a conformational stretching state.^{40,41} The stretching motion during the helicity interconversion would be more restricted within the confined interlocking helicate than within the monomeric helicate. Thus, the activation barrier of the P/M helical interconversion is higher for the dimeric helicate, and their isomer exchange is slow on NMR time scale. Encapsulation of Cl^- anions to the dimeric helicate would further compress the dimeric helicate, and the stretching motion is thus more restricted in this condition. Moreover, the barrier becomes higher within the dimeric helicate may also due to the hindrance that arises from the adjacent aryl groups in the crossed sections, in other words, because aryl–aryl interactions must be disrupted to allow amide twisting.

The discussion and the relevant references (39, 40) has been added and clarified.

It is strange that the chloride-free 4S exists as the equimolar mixture of four diastereomers (PPPP, MMMM, PMMP, and MPPM) (Fig. 2e), although their structures are so crowded that the P/M conversion cannot be observed by means of NMR. Why is the chirality preference not observed at all from 1H NMR integral ratio at this stage?

Response: We thank the reviewer for pointing this out. In the structure of the dimeric helicate **4S**, we could see four diastereomers (PPPP, MMMM, PMMP, and MPPM) out of ten theoretical diastereomers (PPPP, MMMM, PPPM, MMMP, PPMP, MPPM, PMMP, MPPM, PPMM, and PMPM). This chirality selection is similar to that of compound **2** without chiral side chains. We inferred that there is still a certain distance between the chiral side chains as well as between them and helical backbone to prevent them from communicating at this stage, so it cannot give the overall structure with chiral induction by the chiral side chains. However, upon encapsulation of Cl^- anions, the structure of interpenetrated helicate must be shortened and contracted by the electrostatic

attractions between Cl^- and Pd^{2+} , and the corresponding proximity between the chiral side chains is closer enough to communicate and can induce the further chirality preference, resulting in one of ten possible diastereoisomers of the penetrated helicate is finally selected. The discussion has been added and clarified.

As the authors admit, the structure of chloride-bound Pd4L8 determined by XRD is not largely twisted (page 11 in the rebuttal letter). Then why does the Cl^- shift the equilibrium toward one helical isomer? What is the structural mechanism? Figure 5d is too abstract.

Response: The performance of chiral communication is based on the spatial approach of the sequences. Upon the filling of Cl^- anions, the relative motion between the two single helices occurs, and the overall structure of the dimeric helicate is more compact. And because of the shortening of the dimeric helicate architecture, the largest aromatic overlapping area of ligands moved to the segments between the inner pyridine rings of one monomeric helicate and the outer phenyl rings of the other crossed helicate, as observed from crystal structures. The chemical shift data of NMR also support this structural change. Further elucidation of the conformations (between *PPPP* and *PMMP*) of imitative dimeric helicites $[\text{2Cl} \llcorner \text{2}]$ shown the aromatic stacking of *PPPP* isomer was more effective than that of *PMMP* isomer, which may explain the chiral preference (Supplementary Fig. S27). We agree that the Figure 5d is too abstract, and we deleted it.

Figure S27. a) Side view, and b) front view of structural imitation of sectional $[\text{2Cl} \llcorner \text{2}]$ with *PPPP* and *PMMP* conformations. The fake structures $[\text{2Cl} \llcorner \text{2}]$ was fabricated based on the modelling of **2**. The results shown the aromatic stacking of *PPPP* isomer was more effective than that of *PMMP* isomer.

The resubmitted manuscript does not change substantially from the initial one. The main story of this paper (Fig. 3) is not supported by the experiments. The evaluation remains unchanged, and the paper is not recommended for publication in Nature Communications.

Response: The main idea of the manuscript describes the control of chirality communication within a series of oligoamide sequences in the hierarchical manner. The chirality is stemmed from the swing of the amide moieties of the sequences, and the *P/M* helical conformation interconversions are highly dependent on the relative spatial distances of the sequences. Our story revealed that three sequential controls ((i) coordination, (ii) concentration, and (iii) ion stimulus) can progressively

regulate the chirality communication between the sequences, as shown in Figure 3.

(i) coordination

The most important data came from NMR of the Pt-coordinated monomeric helicate, which shows two sets of peaks and their ratio can be biased and regulated by adding different anion guests. This phenomenon cannot attribute to desymmetry of the complex structure on itself, but undoubtedly indicates that there are two isomers (because of *P/M* helicity) on slower exchange in NMR timescale for the monomeric helicate. The result also supports our hypothesis that the metal coordination can correlate and unify the helicity of the ligands within a helicate structure, because there would be more complicate NMR signals for the helicate if the twisting of each ligands was not consistent.

(ii) concentration

Concentration converts the monomeric helicate to the dimeric helicate. The NMR data of the dimeric helicate show that two pairs out of ten isomers (four pairs of enantiomers including *PPPP/MMMM*, *PPPM/MMMP*, *PPMP/MMPM*, and *PMMP/MPPM*, two meso compounds *PPMM*, and *PMPM*) are selected. We posited that the aromatic stacking plays a role in this chirality selection for dimerization. Molecular modelling showed the two pairs of the interpenetrated helicate enantiomers (*PPPP/MMMM*, and *PMMP/MPPM*) were the most energetically-minimized structures, as the tight packing of the aromatic rings could be observed in these conformations. This π - π stacking was also supported by NOE experiments, which showed that the corresponding protons shifted to upfield. In this way, the regulation of chirality communication can be realized by use of changing concentration.

(iii) ion stimulus

Upon encapsulation of chloride ions, only one pair of enantiomers (*PPPP* and *MMMM*) of the dimeric helicate [2Cl^-]**2** was selected, as supported by X-ray crystal structures and the NMR data showing the number of signals of [2Cl^-]**2** reduced by half compared to that of **2**. This chiral preference can also be explained from the perspective of aromatic stacking interaction and elucidated by the conformations (between *PPPP* and *PMMP*) of imitative dimeric helicates [2Cl^-]**2**, which showed the aromatic stacking of *PPPP* isomers was more effective than that of *PMMP* isomers. Upon encapsulation of chloride ions, the largest aromatic overlapping area of ligands moved to the segments between the inner pyridine rings of one monomeric helicate and the outer phenyl rings of the other crossed helicate due to the contraction of the helicate, which was supported by the crystal structures and the chemical shift data of NMR.

In addition, the contraction motion of the dimeric helicate upon encapsulation of chloride ions can be reflected by the CD experiments of **4S** with chiral side chain: the CD intensity is largely increased with encapsulation of chloride ions, since only at this stage the proximity between the chiral side chains as well as between them and helical skeleton is close enough to communicate and induce the chirality preference.

In summary, we have revised the manuscript and sorted out all our data analysis including new experiments, discussions and references to support the main idea shown in the figure 3. We now hope that the resulting work may be found suitable for publication in Nature Communications.

REVIEWERS' COMMENTS

Reviewer #3 (Remarks to the Author):

The authors revised properly according to my comments. Thus, I recommend this manuscript for the publication in Nature Communications.

Reviewers' comments:

Reviewer #3:

The authors revised properly according to my comments. Thus, I recommend this manuscript for the publication in Nature Communications

Response: We are very grateful to the reviewer for his recognition of our research and for his valuable comments.